# Effects of New Zealand Blackcurrant Extract on Sequential Performance Testing in Male Rugby Union Players

**DOI:** 10.3390/sports10100152

**Published:** 2022-10-12

**Authors:** Patrick J. M. Burnett, Mark E. T. Willems

**Affiliations:** Institute of Sport, Nursing and Allied Health, University of Chichester, College Lane, Chichester PO19 6PE, West Sussex, UK

**Keywords:** polyphenols, anthocyanins, exercise, sport

## Abstract

Previous studies on performance effects by New Zealand blackcurrant (NZBC) extract used mainly a single exercise task. We examined the effects of NZBC extract in a battery of rugby union–specific tests including speed, agility and strength testing. University male rugby union players (*n* = 13, age: 21 ± 2 years, height: 182 ± 6 cm, body mass: 87 ± 13 kg) completed two full familiarisations and two experimental visits in an indoor facility. The study had a double blind, placebo-controlled, randomised, crossover design. For the experimental visits, participants consumed NZBC extract (210 mg/day of anthocyanins for 7 days) or placebo with a 7-day wash-out. Testing order was the running-based anaerobic sprint test, the Illinois agility test, seated medicine ball (3 kg) throw, and handgrip strength. With NZBC extract, there may have been an effect for average sprint time to be faster by 1.7% (placebo: 5.947 ± 0.538 s, NZBC extract: 5.846 ± 0.571 s, d = −0.18 (trivial), *p* = 0.06). However, with NZBC extract there may have been reduced slowing of sprint 2 (d = −0.59 (moderate), *p* = 0.06) and reduced slowing for sprint 6 (d = −0.56 (moderate), *p* = 0.03). In the Illinois agility test, there may have also been an effect for the mean time to be faster by 1.6% (placebo: 18.46 ± 1.44 s, NZBC extract: 18.15 ± 1.22 s, d = −0.24 (small), *p* = 0.07). The correlation between the %change in average sprint time and %change in mean agility time was not significant (Pearson R^2^ = 0.0698, *p* = 0.383). There were no differences for the seated medicine ball throw distance (*p* = 0.106) and handgrip strength (*p* = 0.709). Intake of NZBC extract in rugby union players seems to improve tasks that require maximal speed and agility but not muscle strength. NZBC blackcurrant extract may be able to enhance exercise performance in team sports that require repeated movements with high intensity and horizontal change of body position without affecting muscle strength.

## 1. Introduction

Rugby union is a field-based contact sport with two 15-player teams competing in 40-minute halves with a 10-minute half-time rest period. In competitive play, regular stops occur due to rule infringements, try scoring and balls going out of play. Performance analysis of rugby union games provided information on the nature of the intermittent high-intensity activity involving forward sprinting, tackling, scrums and static holds [1]. Agility [2], muscular strength and power [3] and anaerobic endurance [4] are key performance requirements for rugby union players. Agility is required to allow evasive actions as break and avoid tackles are an offensive strategy for field positioning [5,6]. Muscular strength and power are required for tackling and withstand in-game contact collisions [7]. Anaerobic endurance refers to the ability to perform repeated sprints with manipulation of speed to correlate with game performance statistics such as tackle breaks and tries scores [8]. Pre-season, rugby union players develop the physical requisites for competition through physical training modalities with aerobic and anaerobic physical conditioning programmes [9,10]. In season, however, the level of physical conditioning training can be difficult to maintain, and the competitive demands require strategies to enhance recovery and competition [11].

Research on the effects of nutritional ergogenic aids has recently addressed the potential application of the use of functional food ingredients to benefit sports performance and recovery. The New Zealand blackcurrant is a berry rich in the polyphenol anthocyanin from the flavonoid family [12]. Health effects by intake of anthocyanin-rich food are linked with vasodilatory, anti-inflammatory and anti-oxidant properties with application for sport and exercise nutrition [13]. Several studies provided evidence for performance enhancing effects by intake of NZBC extract but the potential to enhance agility has never been examined. In addition, most studies examined the effects on a single exercise task. For example, Perkins et al. [14] observed enhanced performance for a high-intensity intermittent treadmill running protocol with an increase in total running distance by 10.8%. In Godwin et al. [15], trained youth academy footballers (*n* = 9) responded to the intake of NZBC extract with reduced slowing during the running-based anaerobic sprint test compared to university team footballers. It was suggested that enhanced peripheral blood flow alongside venous return influence recovery [14]; therefore, NZBC extract could improve recovery from high intensity activity which could improve repeated sprinting, acceleration and static exertion performance. Deterioration of repeated action often occurs following physical fatigue [16], and NZBC extract seems to postpone or reduce mechanisms of peripheral fatigue during high-intensity and sprint running [14,15], abilities that are important for rugby union players. No studies have addressed in rugby union players the potential of NZBC extract to affect multiple and repeated task performance.

Therefore, the aim of the present study was to examine the effects of the intake of New Zealand blackcurrant extract on rugby union–specific performance tasks, including sprint running, agility, upper-body strength and power and handgrip strength.

## 2. Materials and Methods

### 2.1. Participants and Ethical Approval

Thirteen healthy male participants (age: 21 ± 2 yrs, height: 182 ± 6 cm, body mass: 87 ± 13 kg) of the University of Chichester men’s rugby union team volunteered to participate in the study. Participants played also rugby at club level with six representing home counties under 21 teams. Participants provided informed consent and completed a health history questionnaire and a food frequency questionnaire with anthocyanin-containing foods and drinks listed in the Phenol-Explorer database [17] to estimate habitual daily anthocyanin intake (45 ± 35 mg of anthocyanins/day). Participants were instructed to maintain their habitual dietary intake. In addition, participants completed a COVID-19 symptom form in the 14 days prior to study visits. Ethical Approval was obtained according to the University Research Ethics Policy (code: 1802797). Participants were instructed to abstain from other dietary supplements during the study. In the 24 h prior to testing, participants were requested to avoid strenuous exercise and abstain from alcohol and caffeine consumption.

### 2.2. Experimental Design

Participants completed four visits. Because only three participants completed the Illinois agility test before, visits one and two were full familiarisation sessions allowing instruction for correct completion of all physical tests in subsequent order, i.e., the running-based anaerobic sprint test, the Illinois agility test, the seated medicine ball throw test and handgrip strength test. In the first familiarisation session, height was recorded using a stadiometer (Holtain Ltd., Crymych, UK) and body mass with digital scales (Seca model 873, Seca Ltd., Birmingham, UK). Following the familiarisation sessions, participants completed two experimental testing sessions. The two experimental testing sessions had a placebo-controlled, randomised, double-blind, cross-over design (see below for the dosing strategy). All sessions consisted of a standardised warm up including passive stretching as reported for rugby union players [18]. For the experimental sessions, participants were instructed to wake up at their normal time and have for breakfast water and one piece of buttered bread.

### 2.3. Supplementation

For each of the experimental sessions, the dosing strategy consisted of intake of capsulated placebo (PLA) or NZBC extract for 7 days. For the NZBC extract condition two capsules were taken every day prior to breakfast with the final two capsules on the day 2 h before testing with breakfast. The dosing strategy was similar to previous studies (e.g., [19,20,21]). The daily intake was 600 mg of NZBC extract containing 210 mg of anthocyanins (CurraNZ™, Health Currency Ltd., Camberley, UK). PLA consisted of 600 mg microcrystalline cellulose M102 with intake following the same protocol as the NZBC extract condition. Between experimental visits 3 and 4 was a 7-day wash-out period. Participants were requested to keep a 48 h food diary before the first experimental visit and were instructed to replicate this food diary leading up to the subsequent experimental visit. Upon the completion of the two experimental visits, participants were requested to identify which visit they believed was the NZBC extract condition and comment on if they felt if there was any difference between the sessions.

### 2.4. Running-Based Anaerobic Sprint Test (RAST)

The RAST consists of 6 sprints of 35 m with 10 s rest between each sprint [22]. Participants completed the RAST by sprint running between two sets of timing gates (Fusionsport SmartspeedTM System (accurate to three decimal places), Fusion Sport, Nottingham, UK) at 35 m apart. On completion of each sprint, there was a run-off of 10 m. The RAST test was implemented to assess the fastest sprint time and average sprint time. The RAST was completed first with the aim to induce a match-fatigued condition with the requirement to perform subsequent physical tasks.

### 2.5. The Illinois Agility Test

Three Illinois agility tests were initiated 1 min after the RAST with a 2 min rest between repetitions [23]. The distances and procedures were taken from the Illinois agility test described by Raya et al. [24], with disc cones for marking of the course. Performance of the Illinois agility test was timed with a stopwatch (accurate to two decimal places) and implemented to assess the fastest time and average time.

### 2.6. Seated Medicine Ball Throw Test

The seated medicine ball throw was conducted to measure upper body arm strength and explosive power according to protocol and instructions in Beckham et al. [25]. In the present study, however, participants sat with their back and shoulders against a wall. Participants threw a 3 kg medicine ball as far they as possible in a forward direction with unrestricted launch angle. The test was completed three times with the distance measured using a tape measure that ran from the wall in the direction of the throw. The test was completed three times with one minute between each throw.

### 2.7. Handgrip Strength

For the measurement of handgrip strength, a digital grip dynamometer (T.K.K 5401 GRIP D, Takei Scientific Instruments Co., Ltd., Niigata, Japan) was used with recordings for both hands. Participants held the dynamometer above their head squeezing as forcefully as possible until their hand was moved down by their side according to the procedure in Quarrie and Wilson [26]. The permitted movement during the handgrip strength test was decided as rugby play requires force production during movement. Participants were given a 1 min rest between each repetition for each arm, with three repetitions being conducted on each arm and recordings subsequently averaged for both arms.

### 2.8. Data and Statistical Analysis

Data calculations and statistical analyses were completed using Graphpad Prism 5 for Windows (Graphpad Software, San Diego, CA, USA). For the RAST and the Illinois agility test, the mean time for each of the tests was calculated. For the seated medicine ball throw test, the mean distance was calculated. For the handgrip strength testing, the mean force of three attempts with right and left arm was taken. Two-tailed paired sampled *t*-tests were used to assess condition effects on mean sprint time and maximal sprint time of the RAST, the mean time for the Illinois agility test, the distance for the medicine ball throw test and the mean handgrip force. For each of the sprints 2 to 6 of the RAST, the slowing for each sprint from sprint 1 was tested with a two-tailed paired sample *t*-test. Normality was checked with a D’Agostino and Pearson omnibus normality test. Non-normal distribution observations were tested with the Wilcoxon signed-rank test. For significant changes, Cohen’s d effect sizes were calculated and considered trivial (d < 0.2), small (d = 0.2–0.49), moderate (d = 0.5–0.79) and large (d ≥ 0.8), respectively. The sample size of 13 participants was similar and higher to previous studies with observations of running performance effects by NZBC extract (e.g., *n* = 9 [15]; *n* = 13 [27]). Data are reported as mean ± SD and 95% confidence intervals. Significance was accepted at *p* < 0.05. *p*-Values of 0.05 ≤ *p* ≤ 0.1 were interpreted according to guidelines by Curran-Everett and Benos [28] in that there may be a true effect for a difference.

## 3. Results

### 3.1. Running-Based Anaerobic Sprint Test

#### 3.1.1. Fastest 35 m Sprint

There were no differences between placebo and NZBC extract conditions for the fastest 35 m sprint (PLA: 5.382 ± 0.423 s, 95% CI [5.126, 5.638 s]; NZBC extract: 5.350 ± 0.418 s, 95% CI [5.097, 5.602 s], *p* = 0.59). NZBC extract does not allow a faster 35 m sprint in a series of six subsequent 35 m sprints.

#### 3.1.2. Mean 35 m Sprints and Individual Changes

There may have been faster mean 35 m sprints in the NZBC extract condition (PLA: 5.947 ± 0.538 s, 95% CI [5.623, 6.272 s]; NZBC extract: 5.846 ± 0.571 s, 95% CI [5.500, 6.191 s], d = −0.18 (trivial), *p* = 0.06) (Figure 1). NZBC extract seems to be able to allow rugby union players to have faster repeated 35 m sprints, maybe due to lower impact of fatigue mechanisms during repeated maximal exercise.

#### 3.1.3. Change in Sprint Performance Compared to Sprint 1

There may have been reduced slowing of sprint 2 (*p* = 0.06) and reduced slowing for sprint 6 (*p* = 0.03) compared to the sprint time of sprint 1 in the NZBC extract condition [sprint 2, PLA: 0.402 ± 0.312 s, 95% CI [0.213, 0.591 s]; NZBC extract: 0.231 ± 0.270 s, 95% CI [0.068, 0.394 s], d = −0.59 (moderate)] [sprint 6, PLA: 1.037 ± 0.304 s, 95% CI [0.853, 1.220 s]; NZBC extract: 0.841 ± 0.389 s, 95% CI [0.607, 1.074 s], d = −0.56, moderate)] (Figure 2).

### 3.2. Illinois Agility Test

There may have been a faster mean time for the completion of three Illinois agility tests in the NZBC extract condition (PLA: 18.46 ± 1.44 s, 95% CI [17.59, 19.33 s]; NZBC extract: 18.15 ± 1.22 s, 95% CI [17.41, 18.88 s], d = −0.24 (small), *p* = 0.07). NZBC extract seems to be able to allow rugby union players to turn in different directions with enhanced speed. There was no significant correlation between the %change in mean 35 m sprint time and the %change in mean Illinois agility time (R^2^ = 0.07, *p* = 0.38).

### 3.3. Seated Medicine Ball Throw Test

There was no difference for the thrown distance of a 3 kg medicine ball (PLA: 5.57 ± 0.58 m, 95% CI [5.22, 5.92 m]; NZBC extract: 5.47 ± 0.60 m, 95% CI [5.11, 5.83 m], *p* = 0.11). NZBC extract has no effect on upper body strength and power required to throw a 3 kg medicine ball from a seated position.

### 3.4. Handgrip Strength

There was no difference for handgrip strength (PLA: 50.1 ± 8.2 kg, 95% CI [45.1, 55.0 kg]; NZBC extract: 49.7± 8.4 kg, 95% CI [44.6, 54.8 kg], *p* = 0.74). NZBC extract has no effect on handgrip strength when tested following the RAST, agility test and seated medicine ball throw test.

### 3.5. Blinding Assessment

Only eight participants (62%) correctly indicated which visit they believed was the NZBC extract condition with comments on the preceding week, e.g., “exercise activity felt easier this week” and on the day of testing, e.g., “muscles felt less tired following repeated drills”. None of the participants provided a neutral comment noticing no differences between sessions, indicating that five participants commented favourably in the placebo condition.

## 4. Discussion

The present study provided for rugby union players with a 7-day intake of New Zealand blackcurrant an ability to reduce slowing of some subsequent 35 m sprints and an overall faster mean sprint during the running-based anaerobic sprint test. In addition, following the running-based anaerobic sprint test, rugby union players had a tendency to have faster times in the Illinois agility test. However, upper body strength and power and handgrip strength were not affected. It seems that New Zealand blackcurrant extract is effective for dynamic maximal intensity repeated running tasks that also require direction changes and is not effective for strength and power in short-duration tasks. The observations in the present study add to the body of knowledge regarding the potency of New Zealand blackcurrant extract to enhance high-intensity running performance [14,15,27]. The novelty of the present study was the observation that agility was enhanced when participants were likely experiencing peripheral muscle fatigue from the repeated maximal sprints. However, it needs to be noted that the effect sizes (Cohen’s d) for changes in sprint and agility performance for the group were trivial and small, i.e., −0.18 and −0.24, respectively. However, for the slowing of sprint 2 and 6, the effect sizes were both moderate (−0.59 and −0.56). Therefore, it seems there is no consistent performance-enhancing effect by intake of NZBC extract on the repeated sprint performance.

New Zealand blackcurrant seems to diminish the development of peripheral muscle fatigue by high-intensity exercise. High-intensity exercise creates peripheral muscle fatigue through the accumulation of metabolites and by-products of metabolic pathways, reduced energy supply and changes in ionic concentrations [29]. Furthermore, the reduction in intracellular muscle pH (acidosis) through these repeated high-intensity efforts [30] decreases muscle excitability over the muscle cell and T-tubular membranes, whilst extracellular potassium and chloride concentrations and intracellular sodium concentrations rise [31]. Preservation of muscle excitability was suggested to occur through intracellular acidosis which enabled modulation of the voltage-gate chloride channel [32]. NZBC extract intake may delay peripheral muscle fatigue by permitting elevated levels of intracellular acidosis potentially countering unbalanced ion concentrations which would normally decrease the excitability of muscle via muscle cell and T-tubular membranes. However, the exact mechanisms for enhanced sprint running and agility performance in rugby union players by intake of NZBC extract remains open for interpretation. Alongside muscular events that may contribute to the delay in peripheral fatigue, the potential of NZBC extract to enhance blood flow [24] may have contributed to the enhanced sprint running and agility performance. It was suggested by Perkins et al. [14] for the development of fatigue during high-intensity intermittent running that potential enhanced blood flow may reduce phosphocreatine degradation and accumulation of metabolites also delaying the onset of fatigue and improving muscular recovery. However, Potter et al. [33] suggested that the effects of NZBC extract in a study on sports climbing could have been psychophysiological opposed to solely physiological. Evidence for this within the present study may be supported by the comments from eight of the thirteen participants who correctly identified the NZBC extract condition and reports of feeling less fatigued in the NZBC session. Interestingly, Gibson et al. [34] reported in rugby players for a blackcurrant-based drink with pine bark and L-theanine a reduction in mental fatigue in players expecting to be in a physically fatigued state from an evening training session. In addition, Watson et al. [35] indeed highlighted a psychological effect following anthocyanin consumption with blackcurrant only by an improvement in cognitive task performance and reduction in monoamine oxidase B levels. It was thought that impediment of monoamine oxidase B would prevent the breakdown of dopamine, which is the neurotransmitter responsible for stimulation of motor performance [35], although a new role for the synthesis of gamma-aminobutyric acid has been evidenced but shown in rats [36]. Interestingly, exercise motivation in mouse was linked with gamma-aminobutyric acid signalling in brain striatum [37]. It is not known whether anthocyanin intake may enhance performance by both the synthesis of gamma-aminobutyric acid and gamma-aminobutyric acid signalling. Future research should address the potential central mechanisms that may enhance high-intensity repeated exercise by intake of New Zealand blackcurrant extract.

A key limitation of the present study was that testing occurred within non-realistic field settings. Rugby union is a field-based contact sport played on grass [38]. Performing on grass surface and simulating contact [39] may have affected the performance tasks in the present study and the effectiveness of the NZBC extract. However, based on the observations of the present study, future work may address the effect of NZBC extract in rugby union match-simulated performance protocol for tasks and game duration. Additionally, we cannot exclude that the order of testing, i.e., the potential fatigue from the repeated sprints and agility test, may have influenced our observations on the handgrip and medicine ball throw tests. Previous work by us has shown that with respect to substrate oxidation, females seem to have been more responsive than males to intake of NZBC extract [25,40]. However, as of now, studies on the effects of NZBC extract on exercise performance have not been conducted with female participants. Finally, although our sample size was equal or higher than previous studies that provided observations on enhanced running performance by NZBC extract [15,27], the present study did not have sufficient power to show significance with paired two-tailed *t*-tests for the mean 35 m sprint time of six sprints and mean Illinois agility time of three tests.

## 5. Conclusions

Intake of New Zealand blackcurrant extract daily (210 mg of anthocyanins) for 7 days provided trivial and small effects on overall repeated sprint performance and agility in rugby union players without effect on upper body strength and power and handgrip strength. However, during the repeated sprints, a moderate effect on slowing was shown for the last 35 m sprint in a series of six.

## Figures and Tables

**Figure 1 sports-10-00152-f001:**
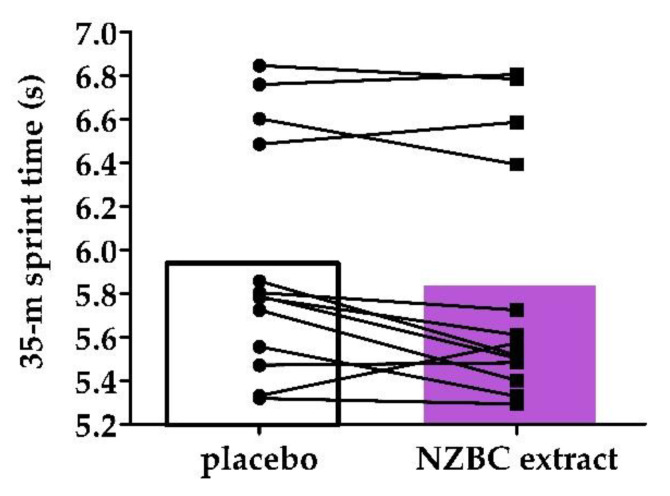
Mean and individual means of the 35 m sprint times for the six sprints in the running-based anaerobic sprint for placebo and New Zealand blackcurrant (NZBC) extract conditions.

**Figure 2 sports-10-00152-f002:**
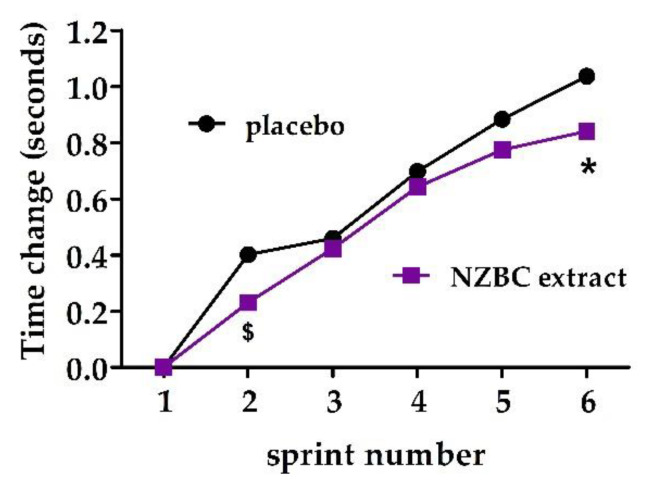
Change in sprint time from sprint one in subsequent sprints in the running-based anaerobic sprint test. $ indicates that there may be a difference between the placebo and New Zealand blackcurrant (NZBC) extract condition (*p* = 0.06). * indicates a difference between the placebo and NZBC extract condition (*p* < 0.05). Data are mean values.

## Data Availability

Data can be provided on reasonable request.

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
