# Peer review of "Effects of New Zealand Blackcurrant Extract on Sequential Performance Testing in Male Rugby Union Players"

_sports, 2022, doi:10.3390/sports10100152_

Round 1
Reviewer 1 Report
You can't say strong trend, this is scientific nonsense. The researchers either need to increase the sample size to ensure the trend is real and avoid false positive data reporting, or report trivial differences only.
"With NZBC extract, there was a strong trend for average sprint time 17 to be lower by 1.7% (placebo: 5.947±0.538 s, NZBC extract: 5.846±0.571 s, p=0.06)"
"In the Illinois agility test, there was also a 19 strong trend for the mean time to be lower by 1.6% (placebo: 18.46±1.44 s, NZBC extract: 20 18.15±1.22 s, p=0.07)"
Also, are the testing regimes sufficiently accurate to detect to 3 decimal points? Why is the average sprint times given to 3 decimal points while the agility test to 2?
Sample size needs justifying. Using a walking study as justification is inadequate.
"food frequency questionnaire to estimate habitual anthocyanin intake 81 (45±35 mg of anthocyanins)." How was this data calculated? What was the FFQ used?
A reader needs more information on the background training status of the participants? Where they club level athletes? Did they compete often? What is their background experience with the various tests?
Why was the hand grip strength test completed by participants with the hand dynamometer above their head? Is this a standardised rugby approach?
Line 269, if you are making an argument of a cognitive effect of BC, you really should be citing the manuscript on BC effects in rugby league players.
"Gibson, N., Baker, D., Sharples, A., & Braakhuis, A. (2020). Improving Mental Performance in an Athletic Population with the Use of Ä€repa®, a Blackcurrant Based Nootropic Drink: A Randomized Control Trial. Antioxidants, 9(4), 316."
Reviewer 2 Report
The authors present a placebo controlled double-blind cross-over intervention study looking at the effect of NZBC on some rugby-specific tests. It's a simple but sound research design however I have some queries relating to the methodology.
The sample is an n=13. I did not see any reference to any statistical power test being conducted a priori to determine the sample size required. Given that the couple of results that the authors deem as being present ("strong trends") I'm wondering whether the study is underpowered to actually determine an effect and whether an increase in participant numbers - as guided by a statistical power test - may yield more definitive findings.
Secondly I am interested in the reliability and repeatability of the chosen performance tests? What is the CV for these tests?
Also re the methods given that this is a nutritional intervention I am curious as to the dietary control of the participants - were there any dietary abstentions required (eg foods high in anthocyanins) as these would effect any intervention.
Also wit regards the statistical methods - I am concerned that repeated T-Tests have been used for the RAST data. These are repeated measures and as such an ANOVA would be more appropriate to reduce type 1 error.
Some other comments:
The writing requires further proof reading, there are numerous grammatical errors and some sentences appear to have words missing.
In the results and also in the discussion you state that 8 guessed the trial correctly (this isn't actually a good thing as it potentially negates the effectiveness of your placebo trial) but you also state comments about 'feeling more energetic etc' however you do not include any statements from the 5 who guessed incorrectly and so there is an element of confirmation bias in what is presented!
You also seem to use the words strength, power and force somewhat interchangeably throughout when referring to the Med Ball throw.
The last sentence of the intro should be removed - this statement appears to be a statement of your results/conclusion (whereas the intro should be written as if the research has yet to be conducted). If this statement is your hypothesis then it needs to be reworded such.
I think that this is a well controlled neat study but I would like the above comments addressed in order to strengthen the manuscript and the outcomes.
Reviewer 3 Report
Interesting work. The presentation was done thoughtfully. One thing I want to caution the authors on is the use of the word "trend". At times it is presented as not so. I hope these comments help:
In the abstract, the word “strong” before trend
In text citations are inconsistent
The mention of caffeine in the introduction is not needed/relevant. It can be removed and the paragraph stands well.
Line 82: change to “COVID symptoms” if it was subjective. Sars-Cov-2 is the virus and should only be mentioned if the presence of the virus was measured.
Line 89: Participants completed 4 visits.
Please move the supplementation portion to where it is after the experimental design. Since you are already introducing it in the design, it makes it more cohesive.
Please make the lines colored in figure 2. It is difficult to distinguish the lines, legend, and symbols when it is all black.
While I commend the authors for showing the individual data, it is difficult to see the trend from the groups. Please add overlapping bars with means or add another figure panel.
Please remove the symbol and mention of the trend in the figures. It may be misleading.
